# The Mitigation of Phytopathogens in Wheat under Current and Future Climate Change Scenarios: Next-Generation Microbial Inoculants

Ixchel Campos-Avelar [1], Amelia C. Montoya-Martínez [1], Eber D. Villa-Rodríguez [2], Valeria Valenzuela-Ruiz [1], Marisol Ayala Zepeda [1], Fannie Isela Parra-Cota [3] and Sergio de los Santos Villalobos [1,*]

1   Instituto Tecnológico de Sonora, 5 de Febrero 818, Col. Centro, Cd. Obregón 85000, Mexico
2   Department of Molecular Biology and Genetics, Aarhus University, Nordre Ringgade 1, 8000 Aarhus, Denmark
3   Campo Experimental Norman E. Borlaug-INIFAP, Norman E. Borlaug Km. 12, Cd. Obregón 85000, Mexico
*   Correspondence: sergio.delossantos@itson.edu.mx

**Abstract:** Wheat production worldwide faces numerous challenges linked to climate change, exponential population growth, nutrient depletion in agricultural soils, and the increasing threat of phytopathogen occurrence. The application of beneficial microorganisms is a promising strategy for crop management as it favors nutrient uptake, improves soil fertility, and increases plant resilience. Therefore, this approach facilitates the transition to more sustainable agricultural practices while reducing the dependence on agrochemicals. The valuable beneficial impacts of bioinoculant application include the enrichment of agricultural soils' ecosystems by restoring microbial populations and interactions that have been lost through the years due to decades of intensive agricultural practices and the massive application of pesticides. Furthermore, beneficial microorganisms constitute a remarkable tool for combating biotic threats, specifically fungal pathogens, whose proliferation and emergence are predicted to increase due to global warming. To optimize their beneficial impact, bioinoculant development requires an extensive study of microbial interactions with plants and their surrounding ecosystem, to improve their composition, mode of action, and stability through application. The use of innovative tools, such as omic sciences, facilitates the elucidation of these mechanisms. Finally, bioprospection and bioformulation must be consciously executed to guarantee the application and persistence of adapted microorganisms and/or their bioactive molecules.

**Keywords:** climate change; bioformulation; bioinoculants; microbial interactions; omic sciences

## 1. Introduction

Cereal crops represent a major source of nutrients around the world, providing proteins, carbohydrates, micronutrients, minerals, vitamins, and fats both for human and livestock nourishment [1]. Wheat is the second most important cereal cultivated globally, with almost 800 million tons produced in 2022 [2]. Two main wheat varieties are grown worldwide, *Triticum aestivum* for flour and *T. turgidum* L. var. *durum* for pasta making, with the first variety being the most cultivated (around 90% of global wheat production) [1].

For many decades, there has been a balance between wheat supply and demand thanks to the implementation of intensive agricultural practices during the Green Revolution, which comprised plant breeding, rigorous irrigation, and the use of agrochemicals [3]. However, high production yields are becoming harder to maintain due to the overexploitation of agricultural lands [1]. This is extremely concerning as the global population is expected to reach around 9 billion people by 2050, and food demand will naturally follow, with predicted increases of 35 to 56%, going up to 62% under climate change scenarios [4]. Currently, wheat yield production around the world faces numerous challenges since intensive agricultural practices are no longer sustainable and have caused the loss of soil fertility.

The massive increase in agrochemical application to maintain high yields has proven ineffective because a significant fraction of them is not used by the plant and prevails in the form of pollutants, i.e., for nitrogen, only 30% is used by the crop [5]. Another important consequence of conventional agriculture is the loss of microbial diversity, affecting soil ecology and restricting the beneficial interactions between plants and microorganisms, which are essential to the plant's fitness to resist biotic and abiotic stressors [6].

Abiotic stressors affecting wheat yield and quality around the world include mainly water deficit, frost events, heat shock, high salinity, heavy metals, and ultraviolet radiation, which affect the photosynthesis process, causing shifts in the developmental stages of the plant, thus damaging floral organs and decreasing the number of grains [7,8]. These abiotic threats for crops will continue intensifying with climate change, as projections indicate shifts in temperature and precipitation patterns, extreme weather events, and an increase in drought periods [9]. Indeed, an increase of 1.5 to 2 degrees in the global temperature is estimated to decrease overall crop production yields by around 5% [10]. For wheat in particular, studies indicate that for each degree of global temperature increase, yields will drop by about 7 to 10% [11]. Furthermore, plants employ numerous resources to resist these various constraints, which causes their weakening, making them more vulnerable to the arrival of biotic menaces such as insects, viruses, and fungal pathogens.

Among the biotic stressors, the proliferation of fungal pathogens is the biggest source of huge economic losses around the world, with an estimated 15 to 20% yield losses each year, which translates to hundreds of billions of dollars [12,13]. The effects of fungal infection on crops vary according to the phytopathogen; they can range from the modification of the physiological characteristics of the plant, reducing its growth and grain yield, to the complete spoilage of grains and the production of harmful mycotoxins. Common fungal pathogens affecting wheat crops nowadays include *Puccinia graminis*, *P. striiformis*, and *P. triticina*, causing infections known as rusts (stem rust, leaf rust, and stripe rust, respectively); *Zymoseptoria tritici* and *Parastagonospora nodorum*, which cause blotches; and *Fusarium* sp., which, besides causing head blight/scab, is also able to produce mycotoxins known as trichothecenes [3]. The impact of fungal phytopathogens on wheat production under climate change scenarios requires a great deal of research as diverse mechanisms are still unknown. However, innovative technologies such as omic sciences and bioinformatic analyses represent major tools for the prediction of future trends.

Considering the economic losses these microorganisms cause and the risk they pose to food security, fungal phytopathogens are heavily combatted by the use of chemical fungicides due to their immediate effectiveness [3]. However, the generation of fungal resistance to fungicides has already been reported, and their efficacy may be further compromised due to altered pest dynamics caused by climate change [14]. On the other hand, chemical pesticides represent strong environmental and safety concerns, which have led to the implementation of complementary strategies, such as crop rotation and diversification, drip irrigation, minimum tillage, and the use of resistant wheat varieties [1]. Despite the various strategies set in place to combat fungal pathogens, they still represent a recurrent threat to wheat. Furthermore, as climate change continues, the variation in climatic conditions will favor the occurrence and distribution of fungal phytopathogens, which will translate to an increase in the occurrence of crop diseases, causing lower yield and grain quality. For instance, phytopathogens that were unable to proliferate in wheat may now have optimal conditions for their development and even become emerging risks for new plant species [15]. Examples of emerging fungal diseases include wheat blast caused by *Magnaporthe oryzae* pathotype *triticum* (MoT), which was exclusively a rice phytopathogen in Asia and then evolved to be able to infect wheat [16]. Another important example of an emerging wheat phytopathogen is the spot blotch disease caused by *Bipolaris Sorokiniana* [17]. Thus, is important to constantly monitor fungal species affecting a specific region to timely detect an eventual increase in the population of a recurrent phytopathogen or to alert on the potential emergence of new fungal threats.

Hence, innovative strategies for crop management rely on the study of microbial interactions taking place in agro-ecosystems. This information allows the optimization of ecological processes towards an improvement in wheat growth and resilience. The extensive knowledge of these mechanisms leads to the development of functional biotechnologies that generate beneficial outcomes, equivalent to those expected from chemical additives (i.e., plant protection and growth promotion), to gradually decrease chemical dependence [18].

## 2. Interactions between Plants and Microorganisms under Current and Climate Change Scenarios

The microbiome associated with crops, which is composed of bacteria, fungi, archaea, protists, and viruses, plays a crucial role in ensuring plant resilience to abiotic stresses, conferring resistance to disease, as well as in facilitating nutrient uptake [19]. Exploiting native microbiomes represents an innovative biotechnological strategy for improving plant resistance to the increasing threats linked to climate change and a key approach for migrating to more sustainable agricultural practices. To achieve this, it is important to understand the ecological processes that govern the interactions of plant microorganisms under different environmental conditions [20].

Two main mechanisms are involved during plant interactions with microorganisms: systemic acquired resistance (SAR) and induced systemic resistance (ISR). SAR involves plant responses caused by phytopathogen attack, while ISR is activated during the interaction with beneficial microorganisms and allows the "priming" of the plant to better resist the eventual pathogenic aggression. It is important to highlight that many interaction mechanisms are still unknown, and they may strongly vary due to climate change; thus, it is important to keep investigating these microbial interactions.

### 2.1. Systemic Responses of Plants upon Phytopathogen Perception

During their evolutionary history, plants have developed strategies to defend against a wide range of phytopathogens, including viruses, fungi, bacteria, and insects [21]. The plant systemic response is a complex network of interconnected signaling pathways and regulatory mechanisms that allow them to activate defense responses systemically (throughout the plant); this mechanism is known as SAR [22]. The systemic responses are energetically demanding and usually associated with reduced plant growth [23]. Thus, to balance defense and growth, plants activate SAR upon phytopathogen perception locally (at the infection site) and spread the signals from the affected site through the plant, inducing a state of readiness in unaffected tissues to respond with enhanced resistance to a following secondary biotic stress [22].

Plants perceive microbial phytopathogens through pattern recognition receptors (PRRs) present on their cell surface (Figure 1). These receptors recognize conserved molecules present in phytopathogens (e.g., flagellin and chitin oligomers), known as pathogen-associated molecular patterns (PAMPs) [21]. The detection of PAMPs by PRRs activates pattern-triggered immunity (PTI), which is often sufficient to stop further pathogen ingress in adjacent tissues. However, more evolved phytopathogens can evade PTI by suppressing it via effector protein releasing [23]. When faced with such situations, plants exhibit an alternative defense mechanism called effector-triggered immunity (ETI). ETI is mediated by receptors located in the cell cytoplasm, which are composed of nucleotide-binding and leucine-rich repeat (NB-LRR) domains located after an amino-terminal coiled-coil (CC) or toll, interleukin-1 receptor-like (TIR) domain (Figure 1). The interaction of NB-LRR receptors with phytopathogen effectors triggers a perturbation of the plant host protein, leading to the activation of ETI [23].

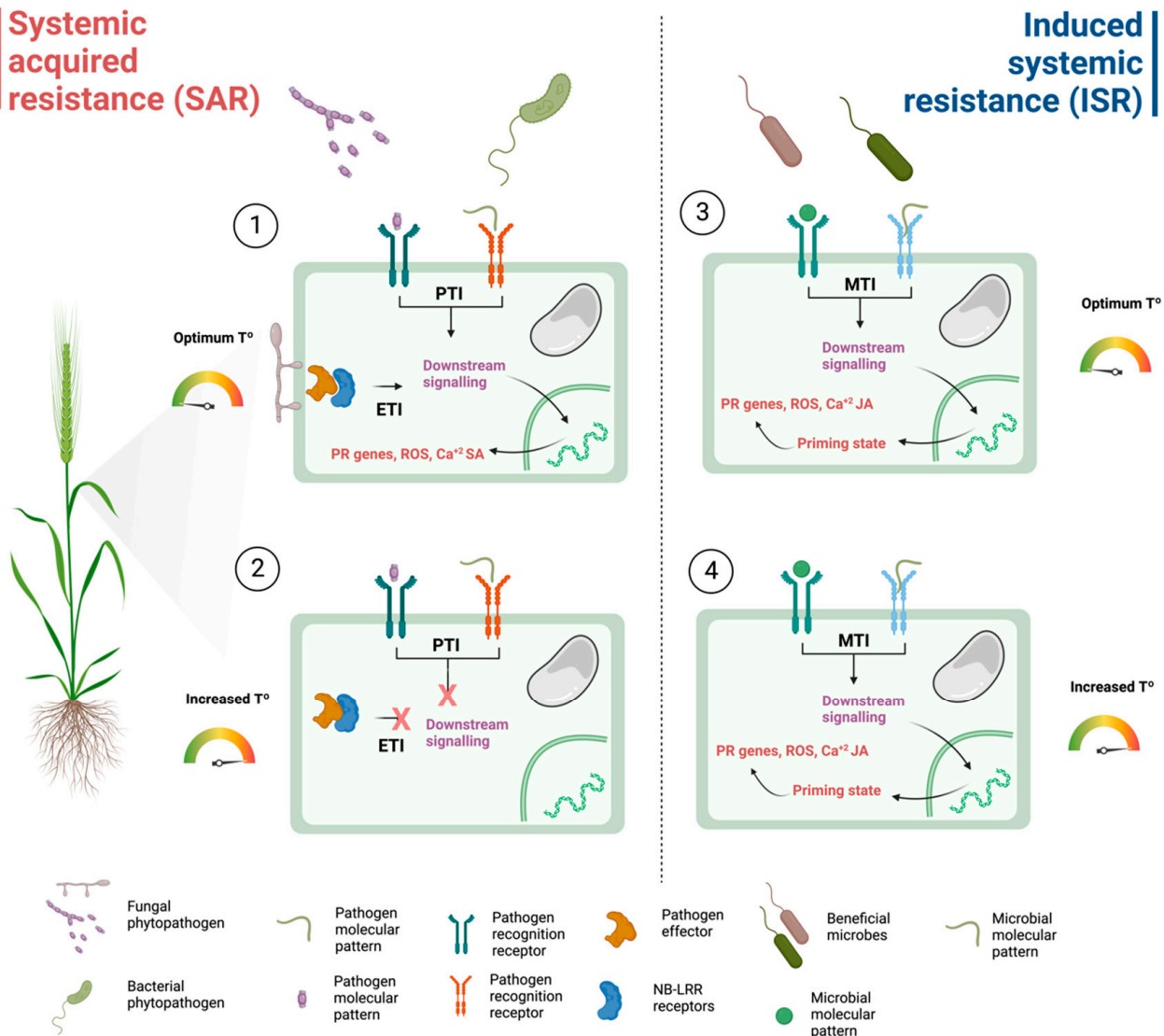

**Figure 1.** Plant systemic defense responses of plants from the perspective of climate change. Systemic acquired resistance (SAR) under optimum ① and ② increased temperature. Induced systemic resistance (ISR) under optimum ③ and ④ increased temperature. PTI (pathogen-triggered immunity). ETI (effector-triggered immunity). MTI (microbe-triggered immunity). PR (pathogen-related). ROS (reactive oxygen species). SA (salicylic acid). JA (jasmonic acid).

Both PTI and ETI lead to downstream events locally, including the production and accumulation of free salicylic acid (SA) and SA-related metabolites, reactive oxygen species (ROS) generation, the activation of mitogen-activated protein kinases (MAPKs), increased levels of intracellular $Ca^{2+}$, the synthesis of phytoalexins, extensive transcriptional reprogramming, and increased expression of defense-associated pathogenesis-related (PR) genes. Subsequently, SA and SA-related metabolites move via the apoplastic route and cuticle to distinct tissues and trigger the activation of defense-related genes. Together, these downstream events allow the plants to stop pathogen ingress/progression at the infection site and distant tissues [23].

Although most research on the systemic defense responses in plants described previously has focused on *Arabidopsis thaliana*, there is evidence to suggest that other economically significant crops, such as cereals, exhibit systemic responses to biotic stress through a similar pathway [21]. For instance, in wheat, SAR-like responses and PR genes have been successfully identified [24]. Furthermore, a recent study evidenced the potential role of jasmonic acid (JA) and SA hormones in wheat's systemic responses [25]. Overall, conver-

gent evolutionary pathways in plants and existing evidence suggest that *Arabidopsis* and wheat have common systemic responses to phytopathogen attacks [26]. However, further research is crucial to comprehensively identify commonalities and distinctions between these two plant species and to elucidate the systemic responses that can be exploited for the optimization of wheat crops.

### 2.2. Plant Defense Responses under Climate Change Scenarios

In nature, plants often experience various biotic and environmental stressors, causing physiological changes that can hinder plant growth and reduce crop yields. The effects of global warming are expected to lead to an increase in heat and drought events, which are already impacting agricultural production and are predicted to worsen in the future [9]. As a result of climate change, plants become more susceptible to phytopathogen attacks due to three primary factors: (i) changes in phytopathogen population dynamics, (ii) evolutionary adaptations of pathogens to evade plant systemic responses, and (iii) disruptions in plant systemic response pathways [9]. Currently, around 13–22% of crop production is lost due to phytopathogen outbreaks, and this is projected to rise due to climate change [27]. In this context, it is crucial to develop alternatives that enhance crop resilience to adapt to changing environmental conditions to ensure food security.

### 2.2.1. Population Dynamics of Native Phytopathogens

The gradual increase in global temperatures has led to a redistribution of soil-borne phytopathogens and changes in population dynamics, such as growth rates, survival, and overwintering patterns. An example of this phenomenon is observed in *Phytophthora infestans*, where higher temperatures have been associated with increased disease severity in potato plants affected by this pathogen [28]. Moreover, experimental studies have demonstrated that even a 1 °C increase during the rice cropping season can substantially elevate the incidence of bacterial panicle blight disease, significantly impacting rice production [29]. For wheat crops, a relevant example is the projected increase in head blight severity in wheat across Europe, which is attributed to shifts in the population of *Fusarium* species. Specifically, *Fusarium culmorum*, which thrives in cool and wet conditions, is being replaced by the more aggressive *Fusarium graminearum*, which prefers warm and humid environments [30].

Plant-associated microorganisms, including phytopathogens, exhibit variations in their physiology and metabolism, which in turn affect their response to changes in the environment. In natural environments, the behavior of these microorganisms is influenced by their interactions with other members of the microbiome (e.g., cooperation and competition) [6]. A recent study demonstrated this phenomenon by isolating naturally occurring phytopathogens from asymptomatic plants and reintroducing them in the absence of the microbial community. As a result, these phytopathogens adopted their pathogenic lifestyle [31]. Consequently, heat events can have a profound impact on the assembly of the microbiome, leading to a disruption in its balance and ultimately causing an outbreak of phytopathogens [6]. For instance, the outbreak of *Bipolaris sorokiniana*, a naturally occurring fungus with a low pathogenic incidence in northwest Mexico, provides an illustrative example. In 2016, this fungus caused spot blotch disease in wheat over a large area of the region, which was potentially linked to the observed rise in temperatures (+2 °C) [32]. Likewise, there is a growing body of literature documenting the increase in unusual disease outbreaks worldwide, including 142 first reports of new diseases, possibly linked to microbiome dysbiosis and phytopathogen redistribution [33].

Although plants have developed SAR mechanisms to detect and combat phytopathogen infections, the ultimate result of this interaction (outbreak or resistance) depends on the disease triangle comprising three key factors: (i) host genetics, (ii) phytopathogen virulence, and (iii) environmental conditions [34]. Traditionally, plant geneticists have focused their efforts on developing resistant cultivars through plant breeding programs to combat recurring diseases. Nonetheless, it is important to recognize that the effectiveness of such

cultivars cannot be guaranteed against the emergence of new diseases caused by the redistribution or emergence of pre-existing phytopathogens in the agricultural system. In this context, it is essential to conduct research aimed at diagnosing the vulnerability of economically important crops. This can be achieved through (i) screening the presence of potential phytopathogens in the region using amplicon sequencing or shotgun metagenomics, and (ii) conducting susceptibility tests under simulated environmental conditions in perspective of climate change (+2 or +5 °C). By employing these strategies, it is possible to gain insights to assist the development of integrated strategies to combat the emergence of new diseases in advance.

### 2.2.2. Phytopathogen Evolution

Plants and microbial phytopathogens (e.g., fungi, bacteria, and viruses) display significant differences that contribute to genetic variation for environmental adaptation. Microbial phytopathogens, due to factors such as genome size, reproduction rate, and population size, can accumulate genetic variation at a faster rate than plants, resulting in a higher evolutionary pace [35]. This imbalance is further intensified within the context of intensive agriculture, where monocultures serve to restrict genetic diversity in plants. The higher evolutionary pace of phytopathogens allows them to develop strategies to evade both plants' PTI and ETI. For example, certain strains of the bacterial wilt pathogen *Ralstonia solanacearum* and *Xanthomonas campestris* have developed nonrecognizable forms of the flg22 sequence (a flagellin epitope), thereby evading the pattern recognition mediated by FLS2 receptor in their respective host plants [36]. Moreover, evasion of effector recognition receptors has been observed by single-nucleotide polymorphism (SNPs), transposon insertions, and deletions of pathogen effector proteins, preventing their recognition by NB-LRR receptors [35]. This is exemplified by the emergence of the rice blast fungus (*Magnaporthe grisea*) as a novel pathogen of wheat promoted by the loss of *PWT3 Avr*, an effector gene that triggers PTI in wheat [37].

While the influence of climate change on phytopathogen evolution remains unclear, previous research has shed light on the impact of increased temperature on their reproduction rate and population density, which can contribute to a faster accumulation of mutations [27,35]. Notably, under heat stress and elevated $CO_2$ conditions, the pathogenicity of *P. infestans* and *C. gloeosporioides* exhibited a rapid increase after months of serial passaging [38,39], evidencing their adaptability to climate change effects, and the impact of these environmental conditions on their virulence. Hence, it is important for future research to assess the impact of mutation rates, promoted by climate change, on the aggressiveness of highly prevalent phytopathogens. Such research will play a pivotal role in determining the vulnerability of crops under projected conditions and help design strategies to enhance their resilience.

### 2.2.3. Systemic Acquired Resistance Signaling under Increased Temperature

Throughout their evolutionary journey, plants have undergone remarkable adaptations that enable them to operate within precise temperature thresholds. As a result, temperature plays a pivotal role in shaping the development and functioning of plants [40]. Notably, temperature influences enzymatic activity within leaves, thereby influencing the developmental growth stage [40]. In this context, it has been observed that plant immune responses are profoundly influenced by temperature and tissue $CO_2$ levels [34,41], which partially explains why plants are more susceptible to phytopathogen attacks under increased temperature. For instance, after being pre-exposed to heat stress, rice resistance to the fungal pathogen *Magnaporthe oryzae* is compromised, exhibiting accelerated tissue necrosis and faster pathogen proliferation [42]. Similarly, under increased $CO_2$ levels and temperature, barley is more susceptible to powdery mildew disease [43].

How temperature and elevated $CO_2$ affect the activation of SAR has been extensively studied in the past years. These environmental conditions can influence SAR functioning at different stages of the signaling pathway. In *Arabidopsis thaliana*, a high temperature reduces

the expression of cell membrane receptor FLS2, involved in flagellin perception, which promotes *Pseudomonas syringae* proliferation [44]. Changes in cytosolic $Ca^{2+}$ concentration are involved in orchestrating downstream immune responses, as observed by attenuated immune responses of defective *Arabidopsis* mutants in (i) *cyclic nucleotide-gated channel 2/4 (cngc2/cngc4)* and (ii) *hyperosmolality-gated $Ca^{2+}$ permeable channel 1.3/1.7 (osca1.3/osca1.7)*. Recently, it has been reported that elevated temperature (28 °C) suppresses cytosolic $Ca^{2+}$ influx in *Arabidopsis* due to low expression of the transcription factors *CBP60g* and *SARD1* [45]. Earlier studies have demonstrated that elevated temperature can also lead to the suppression of ETI (Figure 1). Specifically, higher temperatures have been observed to adversely impact the stability and/or nuclear localization of several disease-resistance proteins, which could negatively affect the pathogen effector recognition. Beyond those, elevated temperature affects other downstream responses of SAR, including ROS generation and SA accumulation [34] (Figure 1). Although it is crucial to acknowledge the influence of climate change on plant immunity, it is important to note that most of the research conducted to date has primarily focused on the plant model *Arabidopsis thaliana*. Therefore, the findings may not necessarily apply to all plant species or plant cultivars due to divergent evolutionary trajectories [34]. In this context, it is important to assess the immune responses of highly valuable crops under simulated environmental conditions from the perspective of climate change (+2 or +5 °C) to evaluate their vulnerability against phytopathogen infections from a climate change perspective [46].

*2.3. Current Management Strategies Employing Plants' ISR*

2.3.1. Systemic Responses of Plants upon Beneficial Microorganisms' Perception

Plants, as with phytopathogenic microorganisms, can perceive and react to beneficial microorganisms similarly by activating microbe-triggered immunity (MTI). Beneficial microorganisms are also perceived by PRRs present in the plant's cell surface (Figure 1). These receptors recognize microbial pattern recognition (MAMPs, synonym of PAMPs), which encompass common structures present in microbes (e.g., flagellin, lipopolysaccharides, peptidoglycan) as well as extracellular metabolites (e.g., siderophores, volatiles, and lipopeptides) [47]. The perception of MAMPs by PRRs can induce plant systemic responses, which results in an ISR phenomenon [48]. Unlike SAR, ISR operates through an SA-independent pathway, where JA and ethylene (ET) serve as the main phytohormone regulators [48]. However, it is worth noting that in certain cases, ISR may require the accumulation of SA [47]. While SAR stimulates a rapid and aggressive response upon perceiving phytopathogens, often leading to hypersensitivity or programmed cell death, microbe-induced ISR initiates a distinct state in plants called "priming" [48]. This state enables plants to generate faster and more robust defense responses when faced with phytopathogen attacks. These responses include the generation of ROS, expression of PR genes, deposition of callose, and production of secondary metabolites (e.g., flavonoids and phenolic compounds) [47]. Due to the broad-spectrum resistance conferred by ISR against various phytopathogens, the use of ISR-inducing microbes or their derived compounds has emerged as a promising biological alternative for managing plant diseases.

2.3.2. Boosting Wheat Immunity: The Use of ISR-Inducing Microorganisms to Mitigate Phytopathogen Impact under Climate Change Scenarios

Ongoing climate change is anticipated to keep exerting a profound impact on worldwide wheat production. With each 1 °C increase in temperature, the yield of wheat could diminish by approximately 1–10% [11]. While it is challenging to precisely determine the specific role of phytopathogens in relation to climate change, their redistribution and rapid evolutionary rate in response to these shifting conditions could potentially lead to devastating consequences for global wheat production [9]. Simultaneously, studies have revealed that certain essential wheat PR genes involved in immunity are sensitive to temperature, indicating that climate change might impede wheat's immune responses [49].

Consequently, it becomes crucial to develop both short and long-term strategies aimed at enhancing wheat's resilience against phytopathogen attacks.

Although SA-dependent systemic responses in plants show temperature sensitivity, as evidenced by phytopathogen resistance and SA accumulation in *Arabidopsis*, an interesting contrast to this is observed in the JA signaling pathway, where the expression of JA response genes is enhanced at elevated temperatures [34,41]. This suggests a potential improvement in ISR under increased temperature. In this context, the use of ISR-inducing microorganisms or their metabolites could be an interesting short-term alternative to mitigate wheat diseases from the perspective of climate change. Indeed, ISR-inducing microorganisms have already shown promising results in controlling wheat phytopathogens (Table 1). For instance, certain bacteria from the wheat microbiome, such as *Bacillus subtilis*, *Bacillus amyloliquefaciens*, and *Paenibacillus* spp. members have been effectively employed to combat wheat diseases caused by *Fusarium graminearum*, *Fusarium moniliforme*, *Gaeumannomyces tritici*, and *Bipolaris sorokiniana* through the induction of systemic defense responses [50–55]. These bacterial genera are known for producing cyclic lipopeptides, such as surfactin and fengycin homologs, which have been proven to trigger ISR responses. In general, wheat treated with such bacteria induces the expression of defense-related genes (e.g., ESD1, Lox2, and PR), induction of phenolic and flavonoid compounds, and callose deposition. Similarly, plant growth-promoting fungi, such as *Trichoderma harzianum*, *Trichoderma asperellum*, *Trichoderma atroviride*, and *Piriformospora indica*, have also demonstrated their effectiveness against wheat diseases (e.g., fusarium head blight and spot blotch) through the induction of ISR [54,56–58]. However, it is important to note that while JA defense responses appear to be unaffected by temperature, as mentioned earlier, much of this research has been conducted in the model plant *Arabidopsis*, and there is limited knowledge regarding their performance in wheat.

**Table 1.** Use of ISR-inducing agents to protect wheat against phytopathogens.

| Wheat Disease | Phytopathogen/Pest | ISR-Inducing Agent | Reference |
|---|---|---|---|
| *Septoria tritici* blotch | *Zymoseptoria tritici* | *Bacillus amyloliquefaciens* | [53] |
| *Septoria tritici* blotch | *Zymoseptoria tritici* | *Bacillus subtilis* lipopeptide mixture (surfactin, fengycin, and mycosubtilin) | [51] |
| Spot blotch | *Bipolaris sorokiniana* | *Trichoderma asperellum* | [58] |
| Fusarium head blight | *Fusarium graminearum* | N-Hydroxypipecolic acid | [59] |
| Fusarium crown rot | *Fusarium graminearum* | *Bacillus pumilis* and *Trichoderma harzianum* | [54] |
| Fusarium head blight | *Fusarium graminearum* | Chitin | [60] |
| Spot blotch | *Bipolaris sorokiniana* | *Bacillus amyloliquefaciens* and *Trichoderma harzianum* | [57] |
| Take-all disease | *Gaeumannomyces tritici* | *Bacillus subtilis* | [61] |
| Sharp eyespot | *Rhizoctonia cerealis* | *Piriformospora indica* | [62] |
| Fusarium head blight | *Fusarium graminearum* | *Piriformospora indica* | [62] |
| Fusarium head blight | *Fusarium moniliforme* | *Serratia marcescens* | [63] |
| The greenbug aphid | *Schizaphis graminum* Rondani | *Bacillus subtilis* | [64] |
| Spot blotch | *Bipolaris sorokiniana* | *Trichoderma harzianum* | [65] |
| - | - | *Bacillus paralicheniformis* | [50] |
| - | - | *Beauveria bassiana* and *Metarhizium brunneum* | [66] |
| Fusarium head blight | *Fusarium graminearum* | *Trichoderma atroviride* | [56] |
| Stripe rust | *Puccinia striiformis* f. sp. *tritici* | *Paenibacillus alvei* | [52] |
| Septoria tritici blotch | *Mycosphaerella graminicola* | *Paenibacillus* sp. | [55] |

Given this context, it becomes essential to thoroughly evaluate ISR-inducing agents in wheat under simulated climate change scenarios. Understanding how these agents respond in wheat will provide valuable insights and help develop strategies to enhance wheat resilience against diseases in the face of climate change.

### 3. Elucidation of Plant and Microorganism Interactions for the Bioprospection of Beneficial Microorganisms

The utilization of omic tools has emerged as a cutting-edge approach for unraveling interaction mechanisms between plants, their environment, phytopathogens, and biological control agents, giving insights into the prediction of future behaviors (Figure 2), as well as the diversity, composition, and function of the plant-associated microbiome [20]. This information is essential for the development of innovative strategies for combating fungal pathogens, improving plant resilience to abiotic stressors, and generally enriching agricultural soils to prevent additional degradation. Furthermore, these technologies facilitate and accelerate the bioprospection of potential biological control agents by enabling the elucidation of numerous pathways from detecting the presence of genes of interest to the identification of the effective production of functional metabolites [67]. Thanks to the advances in omic tools, it is possible to predict the behavior of these interactions under future scenarios and to facilitate the bioprospection for potential beneficial microorganisms that remain active despite environmental changes.

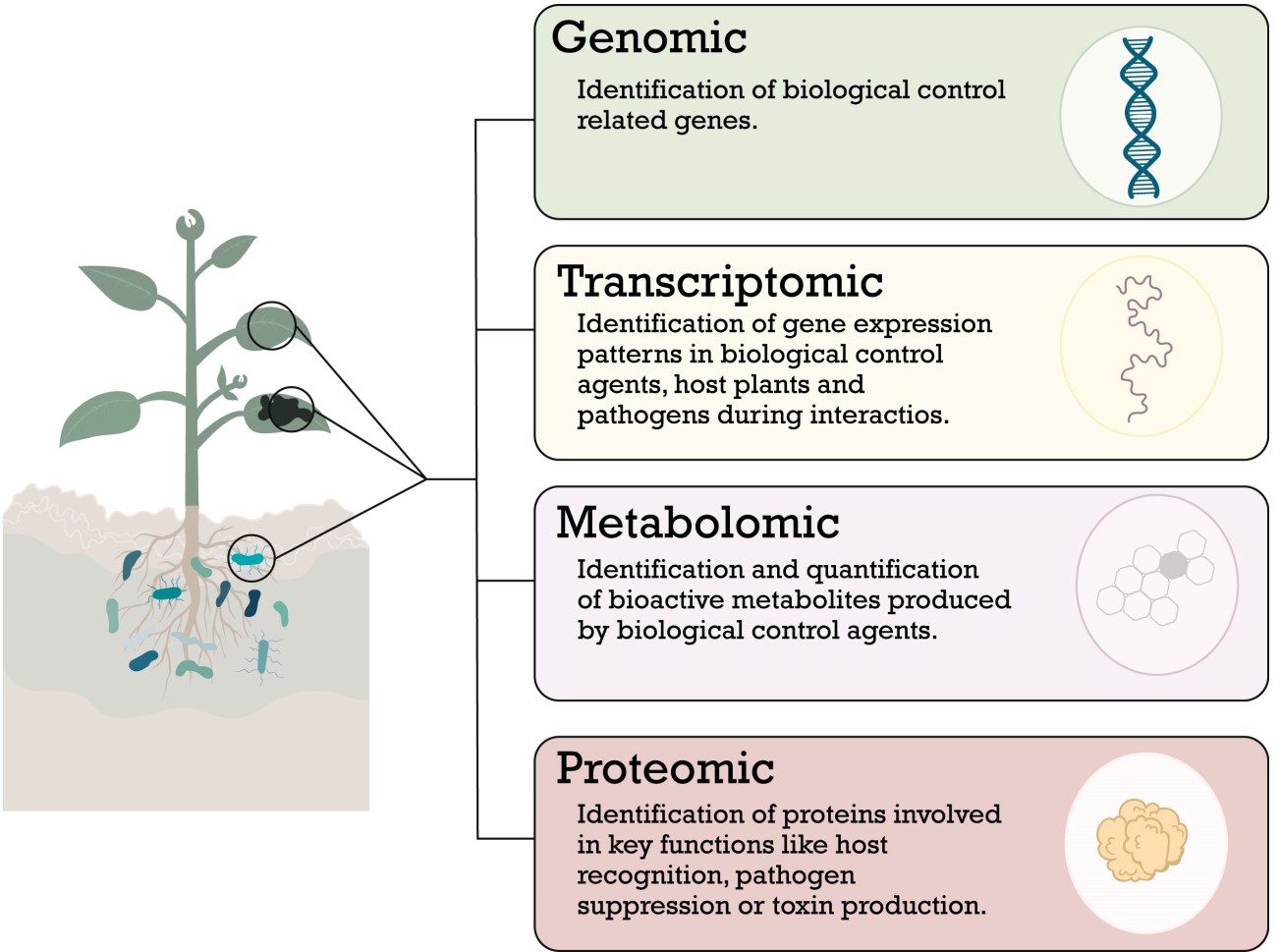

**Figure 2.** Application of omic sciences for the deciphering of mechanisms involved during the interaction of host plant–pathogen–biological control agent.

An example of evolving microbial mechanisms is given by the emergence of phytopathogens, such as *Bipolaris sorokiniana*, the causal agent of spot blotch, which was identified in 2016 in the Yaqui Valley [32], a vital wheat-producing region in Mexico. Research suggests that this pathogen's distribution and disease severity could be significantly influenced by climate change [68]. As climate patterns shift, with projections of increased temperatures and altered precipitation regimes, *B. sorokiniana*'s dynamics and epidemiology may be altered [69]. Higher temperatures and extended periods of humidity create favorable conditions for pathogen growth and disease development, leading to increased infection rates and severity [17]. Changes in precipitation patterns can affect the dissemination of fungal spores, influencing the spatial and temporal distribution of the disease. Spot blotch can cause substantial damage to wheat crops, resulting in reduced grain yield (from 15–25% [70] but may increase up to 85% yield loss [71] under relative humidity > 70% and an average temperature of 20–30 °C) and poor grain quality [72]. To date, many strategies to address the challenges posed by phytopathogens under climate change perspectives, are being implemented such as effective disease management strategies, alongside climate-smart agricultural practices, including the use and application of biological control agents (BCAs) to mitigate the impact of such phytopathogens in a sustainable manner.

The use of synthetic pesticides has long been a conventional approach in traditional agriculture for combating pests and diseases. While these chemical compounds have demonstrated immediate effectiveness in pest control, their widespread use raises significant environmental and health concerns. Synthetic pesticides can lead to the development of pesticide-resistant pests, disrupt ecosystems, and contaminate soil and water resources [73]. Moreover, under climate change scenarios, the efficacy of synthetic pesticides may be compromised due to altered pest dynamics and unpredictable weather patterns [68]. In contrast, the application of BCAs offers a sustainable alternative for pest and disease management. As climate change intensifies, BCAs demonstrate greater potential for adaptation and effectiveness, as they form part of ecosystem-based approaches that promote ecological resilience and stability [74]. Embracing biological control agents in agriculture can not only mitigate the adverse effects of synthetic pesticides but also contribute to sustainable farming practices, fostering ecological balance and ensuring long-term agricultural productivity in the face of a changing climate.

However, available research in understanding microorganism, plant, and soil interactions at a molecular level is limited, especially those focused on integrating omic sciences to perceive the big picture of microorganism function and interactions. The design and formulation of most commercial bioproducts lack profound scientific evidence of action mechanisms and microorganism function, which is crucial to improving their efficacy and predicting their impact on the existing microbiome. The implementation of omic tools has revolutionized our understanding of BCAs and their applications in agriculture [75]. Genomics, transcriptomics, proteomics, and metabolomics have provided valuable insights into the molecular mechanisms underlying BCA-phytopathogen interactions (Figure 2), enabling the identification and characterization of potential BCAs with enhanced biocontrol capabilities [67]. These advanced technologies have facilitated the discovery of novel bioactive compounds produced by BCAs, which play a vital role in suppressing plant pathogens [76,77]. Omic analyses also enable the optimization of BCA formulations for enhanced efficacy and stability under different environmental conditions [78]. Furthermore, metagenomic approaches have revealed the complex interactions between BCAs and the indigenous microbial communities present in the rhizosphere, providing valuable insights into their synergistic effects and potential for ecosystem-based pest management strategies [79]. These advancements in omic tools have significantly contributed to harnessing the full potential of BCAs as sustainable alternatives for integrated pest management and crop protection in agriculture.

In the context of host–pathogen interactions, omic tools have been instrumental in identifying the genes and pathways involved in the plant's defense response against pathogens [75]. Starting with genomic analysis which serves to properly taxonomically

identify a microorganism and provide knowledge on a microorganism's full genomic potential where potential bioactivities of interest may be inferred by gene cluster identification [80,81]. For example, genomic studies have revealed the genetic determinants of pathogenicity in plant pathogens and the genetic basis of biocontrol mechanisms in beneficial microorganisms [82], highlighting recent research under a genomic scope for the elucidation of biological control-related genes utilizing platforms such as the antibiotics and secondary metabolite analysis shell (antiSMASH) [83] post-genomic sequencing, data curation, and genome assembly and annotation [84–89].

Furthermore, transcriptomic analyses have revealed the upregulation of specific defense-related genes in plants upon pathogen attack, shedding light on the molecular mechanisms of plant immunity [90]. For example, the study carried out by Chaparro-Encinas et al. [50] observed the response of wheat from the inoculation of plant growth-promoting bacterium (PGPB) and a BCA, *Bacillus paralicheniformis* TRQ65. The results exhibited the regulation of multidimensional cell growth, suppression of defense mechanisms, induction of central stimuli receptors, carbohydrate metabolism, and phytohormone-related transport, suggesting that *B. paralicheniformis* TRQ65 is a promising bioinoculant agent for increasing wheat growth and development. The suggested mechanism acts by reprogramming ISR and SAR simultaneously, suppressing defense mechanisms and inducing central stimuli response.

Research on transcriptomics has gained recent interest; however, much work is still needed to understand the specific molecular interactions of wheat, which include its response to infection by different phytopathogens and the bioactivity of potential BCAs. Diverse *Bacillus* species have been studied to understand these mechanisms, such as *B. velezensis*, which reportedly up-regulated genes related to sporulation and phosphate stress and down-regulated genes related to secondary metabolism, biofilm formation, and the tricarboxylic acid cycle, when cultivated in the presence of *Fusarium graminearum*, the causal agent of *Fusarium* head blight [91].

Similarly, proteomics enables the identification and quantification of proteins expressed during plant–pathogen and plant-BCA interactions. Proteomic studies have uncovered the key players in the host immune system that respond to pathogen invasion, as well as the bioactive compounds produced by beneficial microbes that inhibit pathogenic growth [92,93]. Integrating proteomic data provides valuable insights into the molecular events that determine the outcome of these interactions, helping to identify new targets for developing biocontrol strategies. These studies have shed light on the role of *Bacillus* biocontrol species in enhancing wheat resilience under climate change scenarios, identifying proteins involved in pathogen recognition and signaling cascades, and contributing to a comprehensive understanding of the plant's defense mechanisms [94–96]. Another example of such research is a study conducted by Abd El-Daim et al. [94], where they investigated the proteomic changes in wheat plants under drought stress inoculated with *Bacillus velezensis* 5113. The researchers found that the application of *B. velezensis* 5113 led to significant changes in the wheat proteome, resulting in the upregulation of proteins related to stress tolerance, antioxidant defense, and photosynthesis. This proteomic response indicated that *Bacillus velezensis* 5113 can enhance wheat's adaptive capacity to cope with drought stress, potentially mitigating the negative impacts of climate change-induced drought on wheat production. Furthermore, *Bacillus velezensis* has been reported as a biological control agent against several wheat diseases, including *Fusarium* head blight [97], take-all, and spot blotch [98].

Also, via a proteomic approach, Rashid et al. [96] explored the impact of *Bacillus megaterium* MU2 on wheat plants under drought-stress conditions. They identified a set of drought-responsive proteins that were significantly influenced by *Bacillus megaterium* MU2 treatment. These proteins were associated with defense regulatory mechanisms and the plant–microbe symbiotic relationship, as well as the formation of cellular spores and phosphorylating activity in the glucose metabolic process. This suggests that *B. megaterium* MU2 can induce a protective response in wheat plants under drought stress, indicating that

the symbiotic interaction between *B. megaterium* and wheat results in a higher tolerance of the plant against drought stress.

Moreover, metabolomics allows the simultaneous analysis of a wide range of small molecules, providing information about the metabolic changes occurring in plants and microbes during their interactions. Metabolomic studies have revealed the production of antimicrobial compounds by biocontrol agents and the altered metabolite profiles in plants upon pathogen attack or biocontrol agent inoculation [95,99]. Integrating metabolomic data helps researchers identify the specific metabolites involved in the defense responses or biocontrol activities, facilitating the development of novel biocontrol agents or bioactive compounds. Metabolomics analyses have highlighted the potential of biocontrol species as sustainable tools for improving wheat productivity and crop resilience under the challenges posed by climate change. For example, the study conducted by Abd El-Daim et al. [94], also explored the metabolomic changes in wheat plants treated with *Bacillus velezensis* 5113. The researchers found that *B. velezensis* 5113 treatment induced significant alterations in the wheat metabolome, associated with the modulation of several metabolic pathways of amino acids, suggesting that *B. velezensis* enhances tolerance of wheat exposed to different abiotic stress factors including heat, cold, and drought.

Promising directions for the future of omic-based biological control strategies lie in harnessing the vast potential of these advanced technologies to further enhance and optimize the efficacy of BCAs in agriculture and pest management. Integrating genomics, transcriptomics, proteomics, and metabolomics approaches can provide a comprehensive understanding of the complex interactions between BCAs and their target pests or pathogens, leading to the discovery of novel bioactive compounds and mechanisms involved in biocontrol [100]. Omic tools can facilitate the identification of key genes and pathways involved in the production of biocontrol-related secondary metabolites, such as antibiotics and enzymes, which are crucial for inhibiting pathogen growth [100]. For example, the research carried out by Villa Rodríguez et al. [67], integrating genomic and metabolomic results from the effect of *Bacillus cabrialesii* TE3$^T$ on *Bipolaris Sorokiniana* TPQ3, revealed that strain TE3$^T$ contains the biosynthetic potential to produce wide spectrum antifungal and antibacterial metabolites, and determined that a lipopeptide complex of surfactin and fengycin homologs was responsible for antifungal activity exhibited by *B. cabrialesii* TE3$^T$ against the studied phytopathogen.

Overall, omic tools have significantly advanced our understanding of interaction mechanisms in various biological contexts. By providing a comprehensive and holistic view of these interactions, omic approaches offer valuable insights that can be harnessed for developing sustainable agricultural practices, biotechnological advancements, and ecological conservation strategies more efficiently. The insights and information obtained using omic approaches elucidate the potential mechanisms of action and molecular pathways of beneficial microorganisms; however, these are not a guarantee that the desired effect will be obtained in a real scenario when applied in the field, as it depends on numerous factors. Furthermore, climate variations impact microbial dynamics as microorganisms adapt, become dormant, or die following environmental changes [101]. Known plant pathogens and effective beneficial microorganisms' mechanisms of action may be strongly modified by the changing climate conditions. Therefore, it is crucial to comprehend current microbial interactions and their potential evolution following climate change as what is valid now may not be in the future. Thus, complex interactions between phytopathogens, beneficial microorganisms, wheat crops, and changing climate necessitate continuous research and proactive management strategies. Understanding how climate change influences microorganisms' biology and evolution is crucial for developing climate-resilient wheat varieties and implementing effective disease control measures. By integrating scientific knowledge, innovation, and sustainable agricultural practices, it is possible to enhance wheat production's resilience under current and future climate change scenarios.

## 4. Development and On-Field Application of Beneficial Microorganisms

When developing bioinoculants, is essential to take into account the resilience of the selected microorganisms during bioprospection, to ensure that their beneficial activities will perdure despite changes in the agroecosystem. This is facilitated by isolating microorganisms from the intended agroecosystem of application, as well as by subjecting potential candidates to tolerance assays to verify if they can resist higher temperatures, lower water activity, etc. Using the information provided by omic analyses will ensure that the selected microorganism has the intrinsic capacity of producing functional activities and metabolites. Indeed, innovative approaches for bioprospection include genome mining for specific genes of interest.

### 4.1. Bioprospection of a Biocontrol Microorganism

The search for novel microorganisms with beneficial traits, particularly bacteria, has emerged as a promising avenue for the development of sustainable and environmentally friendly solutions for plant disease management [102]. Bioprospecting beneficial bacteria to develop a bioinoculant involves a systematic exploration of diverse agroecosystems, and from different substrates such as soil, rhizosphere, or plant tissue, in search of microorganisms with desirable characteristics. The process typically begins with sample collection, followed by isolation and identification of the bacterial strains. Then, a characterization of their biochemical and metabolic properties is carried out. After this, a series of tests and assays are carried out to determine their abiotic stress tolerance, and their expected bioactivity and beneficial traits (i.e., biological control and plant growth promotion). Lastly, the development of products and their commercialization, including trials, patenting, marketing, and sales, is carried out [103]. It is essential to incorporate new technologies and robust tools into the bioprospection of biocontrol bacteria, especially regarding their taxonomic affiliation, mode of action in vitro, and under greenhouse and field conditions, as well as their interaction with biotic and abiotic factors [75]. In this section, the most critical steps of bioprospecting bacteria for biological control purposes will be discussed (Figure 3).

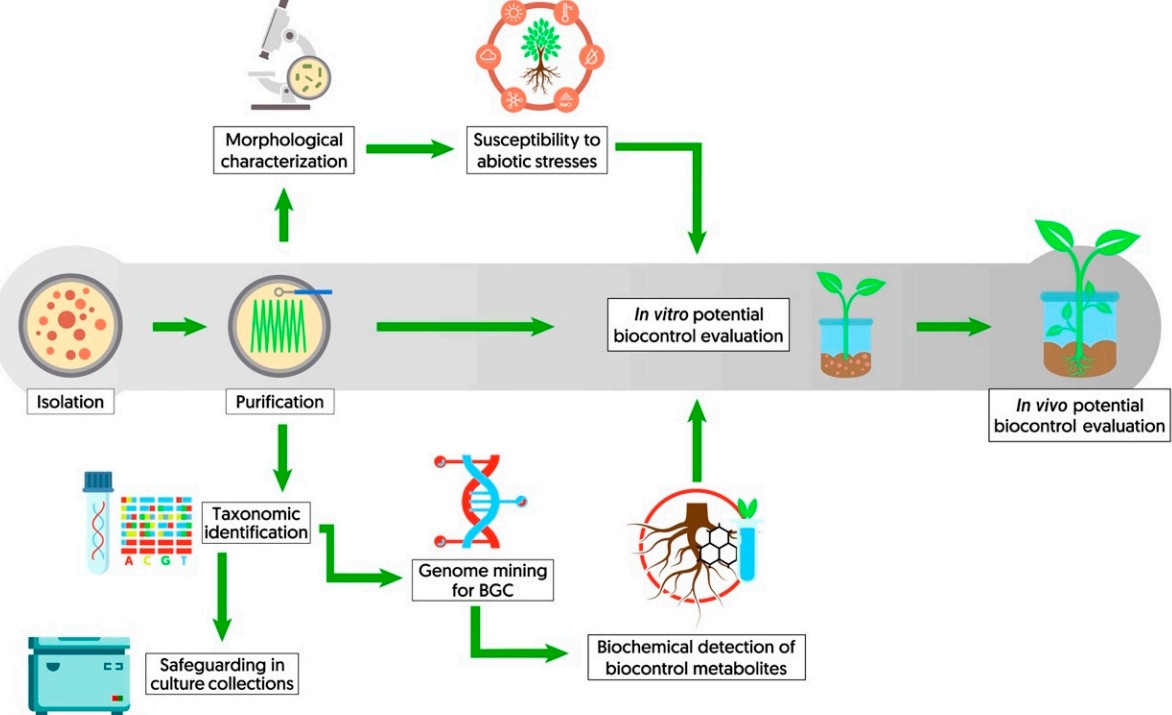

**Figure 3.** Workflow for bioprospecting a biological control agent for its application as a bioinoculant.



The agroecosystem and its environmental conditions play a crucial role in the adaptation and establishment of microorganisms. Therefore, to bioprospect a successful bioinoculant, it is necessary to obtain and isolate said microorganisms from an agroecosystem with similar conditions to those where the bioinoculant will be used. Although BCAs are stable, their use has not been very successful in the field due to the diversity of interacting factors that are exempt under laboratory conditions, and those challenges increase with the ongoing climate change. To overcome this, it is critical to choose agents that are effective under a variety of environmental situations (soil texture, temperature, humidity, radiation) [104]. Although no single screening method is optimal for all biocontrol endeavors, it is essential to adopt a rational approach that aligns with the specific pathosystem (plant–pathogen–environment) of interest; for example, finding BCAs against foliar-specific pathogens would likely require screening microorganisms that can colonize the phyllosphere [105].

Once the isolates are obtained, the identification process provides crucial insights into the taxonomic classification, genetic diversity, and potential functional attributes of the isolated strains. For this, the use of modern sequencing and bioinformatic technologies allows for a precise taxonomy affiliation [75,80]. The use of whole-genome sequencing and assembly also allows for genome mining for biosynthetic gene clusters (BGCs) that potentially produce secondary metabolites for biocontrol [75]. Different bioinformatic tools annotate and predict BGCs, such as PRISM [106], RiPPMiner [107], CLUSEAN [108], ClustScan [109], BAGEL4 [110], and AntiSMASH [111], which is one of the most widely used pieces of software for secondary metabolite gene cluster identification, annotation and analysis, due to its comprehensive, rapid and user-friendly interface [112]. Once the bacterial strains are taxonomically identified, a thorough literature research would help to select those strains from species previously reported to have biocontrol activity, and the genome mining will allow to confirm potential biocontrol activities for the specific strain. The application of these tools has proven efficient for bioprospecting and elucidating the modes of action of potential BCAs. For example, in a recent study, Dutta et al. [113] evaluated the contribution of secondary metabolites produced by *Pseudomonas fluorescens* NBC275 (Pf275) to its antifungal and biocontrol activity, combining in silico analysis of the genome using AntiSMASH with a previous study of transposon (Tn) mutants. They found that the biocontrol performance of Pf275 was dependent on the antibiotics 2,4-diacetyl phloroglucinol and pyoverdine, but also found clusters that encode for aryl polyene and an unidentified small linear lipopeptide that influenced antifungal and biocontrol activities. In another study aimed to obtain information and elucidate the antifungal mechanisms of *B. amyloliquefaciens* strain GKT04 against *Fusarium oxysporum* f. sp. *cubense* race 4 (FOC4), Tian et al. [114] used a combination of genome analysis and transcriptome sequences and found multiple BGC involved in the synthesis of antibiotic metabolites, including polyketides, siderophores, and lipopeptides, of which genes encoding polyketide difficidin, the siderophore bacillibactin, and the lipopeptide bacilysin were found to be upregulated in response to FOC4.

Other approaches can be taken to determine the potential biocontrol activity of a microorganism. For instance, to detect the synthesis of antibiotics in BCAs, HPLC is usually used [115]; lytic enzyme production such as cellulase, chitinase, and glucanase, can be detected by measuring the degradation of specific substrates and confirmed by spectrophotometric methods [116]. Swarming motility and hemolytic activity can be used as pre-indications of lipopeptide production, as many swarming bacteria synthesize and secrete surfactants that reduce tension between the substrate and the bacterial cell and, in doing so, can permit spreading over surfaces [117]; hemolytic activity has been reported to be present in some members of the iturin and surfactin families [115] and can be easily detected by inoculating the potential BCAs on Petri dishes containing blood agar, where hemolytic halos (β-hemolysis) are observed around the colonies after few hours of incubation [17,118].

The detection of BCAs can be accomplished by conducting confrontation assays on Petri dishes. These assays involve placing the potential BCAs (antagonists) in a confronta-

tion with a phytopathogenic bacteria or fungi. The trials can be designed with different combinations, such as one antagonist against one pathogen, and two or even more antagonists against one pathogen [115]. The inoculation method involves applying a drop containing a specific quantity of colony-forming units per milliliter (CFU ml$^{-1}$) or using a microbiological loop of inoculum (antagonist and pathogen) at a certain distance from each other. After the incubation time, the growth of the pathogen colony is measured and compared against a control that consists of a pathogen growing without an antagonist [115,119]. In 2019, Villa-Rodríguez et al. carried out a high-throughput qualitative dual assay in Petri dishes against *Bipolaris sorokiniana* to identify antagonistic bacteria; out of 195 tested bacteria, 6 strains isolated from bulk soil, 7 strains isolated from the rhizosphere, and 1 endophytic strain showed antagonistic activity against *B. sorokiniana* with inhibition zones ranging from 1.6 to 8.0 mm [17], which allowed them to select potential BCAs. Similarly, a confrontation assay conducted on agar demonstrated *B. subtilis* strain PTB185 antagonistic activity against *Botrytis cinerea*, *Pythium ultimum*, *Sclerotinia sclerotiorum*, *Mucor* sp., and *Rhizoctonia solani* [119].

Abiotic stress tolerance is an important attribute in the selection of bacteria for the development of microbial inoculants. After selecting the bacterial strains with the best biocontrol activity *in vitro*, and knowing the potential metabolites involved in said biocontrol, it is necessary to determine the abiotic tolerance of said microorganism. This step will allow for the selection of biocontrol strains that have a better chance of establishing in adverse field conditions, such as high salinity, acidic or alkaline soils, hydric stress, or high temperatures. The assays are usually performed using different concentrations of NaCl, polyethylene glycol, or augmenting the incubation temperature, for salt tolerance, hydric stress tolerance, or temperature tolerance, respectively [96,120,121]. A pH range for optimal growth could be determined by growing the bacteria in acidified or alkalinized media [122,123]. In 2020, Dixit et al. [124] determined the alkalinity tolerance ability of 15 selected PGPB isolates and found that the isolate *Bacillus* sp. NBRI YN4.4 showed maximum tolerance at pH 9 and pH 11; moreover, it significantly enhanced soil enzymes such as dehydrogenase, alkaline phosphatase, and betaglucosidase due to inoculation of NBRI YN4.4 in maize plants grown in alkaline soil [124].

Conducting in vivo biocontrol evaluations in both greenhouse and field settings allows for a reliable assessment of the biocontrol agent's effectiveness [105]. This ensures that these bioinoculants gain the producer's confidence, making them suitable for commercialization and long-term practical application. The specific experiments and tests may vary depending on the particular pathosystem, but their ultimate objective is to develop a successful bioinoculant. It is crucial to note that laboratory efficacy does not always guarantee success in the field. The ecological interactions occurring in the field play a significant role in determining whether biocontrol can be achieved. Bridging the gap between laboratory and field efficacy requires a comprehensive understanding of the in-field mode of action of the bioinoculant to predict the conditions under which it can be effectively applied [125].

The reintroduction of bioinoculants to agricultural lands will mitigate the adverse effects of pesticides and artificial fertilization, restoring symbiotic relationships that may have been downregulated due to chemical application [126], thus reinstating an ecological balance and favoring nutrient uptake by the plant, ensuring long-term agricultural productivity and sustainable farming practices in the face of a changing climate.

However, a crucial step for the application of effective bioinoculants is their optimal bioformulation, to ensure their establishment and optimal development in the field. This process requires extensive knowledge of the stability, compatibility, and versatility of the selected beneficial microorganisms and/or molecules. The main points to consider during the bioformulation process will be further described.

### 4.2. Formulation of Bioinoculants for Field Application

Bioinoculants face numerous challenges during their application in the field, which affects their survival and efficacy. These obstacles may include the environment and

the presence of other microorganisms, the lack of adaptation to the intended site, poor stability of the formulation, or short shelf-life; also, the presence of chemical fertilizers in the field limits the viability of bioinoculants [127]. It is well known that a biopesticide or biofertilizer that exhibits promising properties under in vitro or laboratory conditions often fails to perform in the field. The success of a bioinoculant is dependent on its ability to overcome said challenges, and this is achieved by optimizing not only the selection of an adapted microorganism but also its formulation according to the intended use. Furthermore, bioinoculant commercialization requires long processes of field trials and validation to obtain legal approval for their distribution on the market [128].

Thus, the development of effective bioinoculants comprises several steps, starting with the bioprospection of biologically active microorganisms and/or their metabolites, followed by the selection of the appropriate carrier and then its bioformulation. It is also important to determine the mode of delivery which is usually either by seed coating, soil inoculation, or foliar spray, according to the type of crop and the irrigation system. Subsequently, the final product has to be evaluated for its resilience, stability, shelf-life, efficiency, and environment friendliness, among others [127]. Usually, preliminary evaluations are conducted in the laboratory or under greenhouse conditions and then on experimental fields.

Following bioprospection, which validates plant promotion or biological control properties of the selected beneficial microorganisms, the compatibility with the carrier has to be verified, as well as the retention of the bioactivity and viability after the intended storage time [129]. As mentioned before, microorganisms isolated from extreme environments are more versatile and have dynamic chemical communications with the soil environment. Indeed, the development of a bioinoculant with a microbial strain isolated from the intended site of application increases the probability of survival as it will be adapted to the soil conditions and microbiome [130].

To favorize bioinoculant application and establishment in the intended ecosystem, it is critical to develop an optimized formulation that increases survival probabilities. Beneficial microbes can be formulated either individually or in a consortium, multi-strain formulations are typically more efficient [130]. Bioformulations can be made either solid-based or liquid-based. Liquid-based ones comprise liquid buffers with protectants such as concentrated suspensions in water, oil, or a mixture of both. Solid-based bioformulations consist of carriers that act as binders and dispersants, such as talc, sawdust, granules, chitin, alginate, starch, fly-ash, perlite, biochar, and peat, among others [130]. Carriers must have certain characteristics such as being nontoxic, chemically stable, readily available, inexpensive, being able to maintain humidity if required, and most importantly, being capable of ensuring microbial cell viability during transport and upon arrival to the targeted site [131].Promising bioformulations have been designed to maintain phosphate-solubilizing bacteria, which reportedly increase wheat grain yield by 4–9% [132]. Spray drying resulted in an effective method for the bioformulation of plant growth-promoting *Bacillus* strains for their application on wheat plants [133].

Next-generation formulations have explored different alternatives to microbial cells, such as cell-free extracts or supernatants, employing innovative carriers such as bioencapsulation, biologically active matrices, and nanoparticles [130]. The secondary metabolites of beneficial microorganisms can also be applied directly as adjuvants, additives, biostimulants, protectants, and antimicrobials [134,135]. Bioencapsulation is a promising formulation technique that uses polymeric substances such as alginate, carbohydrate matrices, metallic nanoparticles, etc., which offers protection and survival of the microorganisms once applied. Bioencapsulation is performed in three main steps: (i) the active principle or metabolite is mixed or absorbed in the polymeric matrix; (ii) a liquid solution is dispersed under agitation and solid particles are formed; (iii) solid particles undergo polymerization and physicochemical stabilization [131]. Other bioformulation techniques include wettable powders (WPs), wettable granules (WGs), and water-dispersible granules (WDGs), which get easily dissolved in water for their application [131]. An innovative approach is the use of nanoparticles as carriers, which offers durability, versatility, and a high surface/volume

ratio, this includes Zn and Fe, developing nano-biopesticides. Novel formulation techniques also include the genetic modification of beneficial strains, which can also provide them with resistance to the changes in their environment; mutations are usually performed by UV radiation, gamma rays, and mutagenesis. Synergic activity can also be obtained by formulating beneficial microorganisms with bioactive compounds such as phytohormones, flavonoids, osmoprotectants, organic amendments, compost, algae extracts, humic acid, silicon, and aminoacids [130].

*4.3. Field Application of Bioinoculants for Wheat: Existing Commercial Products and Guidelines*

As mentioned before, the integration of phytomicrobiome members into agro-ecosystems is an innovative technology to mitigate the increased biotic and abiotic stressors predicted by climate change. However, the successful application of bioinoculants in the field requires their optimal bioformulation and correct application [136,137].

Several strains among different genera have been studied for their biocontrol activities against phytopathogenic microorganisms. For example: *Paenibacillus xylanexedens*, *Bacillus* spp., *Azotobacter* spp., *Streptomyces* spp., *Ochrobacttrum intermedium*, *Paenibacillus lentimorbus*, *Trichoderma* spp., *Pseudomonas* spp., *Pythium oligandrum*, *Bacillus velezensis*, *Bacillusamyloliquefaciens*, *Priestia* sp. TSO9 [87–89,138–140]. Nevertheless, only a few species are registered as commercial biofungicides for their use in the control of wheat diseases. Some of them are listed in Table 2.

**Table 2.** Commercial bioproducts based on biological control agents against diseases in wheat.

| Biological Control Agent | Commercial Product | Target Disease in Wheat | Company | Reference |
|---|---|---|---|---|
| *Pseudomonas chlororaphis* strain MA342 | Cerall® | *Fusarium* head blight, *Tilletia tritici* and *Tilletia laevis* wheat bunt | BioAgri AB, Uppsala, Sweden | [141] |
| Chitinolytic activities or by-products of microbial detoxification of mycotoxins. *Trichoderma asperellum* | Xedavir® | *Fusarium verticillioides* | Xeda | [142] |
| Mycophageous action. *Pythium oligandrum* strain M1 | Polyversum® | *Fusarium graminearum* head blight | AgrichemBio | [143] |
| *Streptomyces* spp. Antigerminative compounds | Mycostop® | *Fusarium* spp. and deoxynivalenol (DON) mycotoxins | Lallemand Specialties Inc. | [144] |
| *Streptomyces* spp. | Actinovate® | *Fusarium* spp. and deoxynivalenol (DON) mycotoxins | Actinovate SP | [145] |
| *Bacillus subtilis* strain QST713 | Serenade® ASO | Yellow rust (*Puccinia striiformis*) | Bayer | [146] |
| Clove extract + *Bacillus subtilis* + emulsifiers, conditioners, and diluents | Roya Out® | Leaf rust (*Puccinia triticina*) | Greencorp | [147] |
| *Bacillus* spp. + *Azotobacter* spp. + *Pseudomonas* spp. + plant extracts + conditioners and stabilizers | Best Ultra® F | Leaf rust (*Puccinia triticina*) | Greencorp | [148] |
| *Gliocladium catenulatum* strain J1446 | Prestop® (WG) | Damping off (*Arthrinium sacchari*), Root rot (*Bipolaris sorokiniana*, *Fusarium* spp.) | Lallemand Specialties Inc. | [149] |

To ensure their efficacy, these commercial formulations of BCAs must be applied following important guidelines, as described below [128,150]:

1.  The product must be of good quality (at least $1 \times 10^7$ viable cells $g^{-1}$) and purchased from a reputed supplier, as well as applied according to the recommendations of the dose.
2.  The product must be used for the crop(s) specified on the product label.

3. While inoculating, excess culture should be inoculated, or any remnants/residual culture should be immediately put in grooves of the field so that inoculum microorganisms start interacting with other microbiota in the rhizosphere and begin colonizing the rhizosphere.

4. To achieve major/expected shelf life, the product should be stored in cool places and away from light sources (room temperature 25–28 °C or cooler, depending on the type of microbial product).

5. Direct contact of the product with herbicides/weedicides/pesticides should be avoided.

6. It is important to have detailed information about the strains and ingredients to know the ideal soil conditions for its application. For example, if the soil is highly acidic, the integration of soil amendments (lime or rock phosphate) is recommended.

Moreover, despite following the appropriate considerations, there are various constraints in the application of BCAs based on the cells of beneficial organisms, which are abiotic and soil-related factors (soil and environmental conditions and compatibility with agronomical practices) as well as biotic and host-related factors (microbial competition vs. colonization, taxonomy of the strains, plant–microbe signaling, and host specificity) [136]. For these reasons, an alternative is the formulation of cell-free supernatants (CFSs), which are mixtures derived from broth cultures subjected to processes that allow the removal of cells while maintaining their produced compounds, such as antibiotic antimicrobial metabolites, lytic enzymes, chitinases, cellulases, glucanases, volatile organic compounds, organic acids, phytohormones, exopolysaccharides, and lipopeptides [135,151].

In addition to a correct BCA application, an integrated management plan to control wheat diseases will be necessary. For example, by combining different strategies, such as crop breeding (the development/use of wheat varieties resistant to the specific disease) and sustainable agricultural practices that reduce susceptibility to some pests/herbs (i.e., crop rotation, minimum tillage systems, and/or balanced fertilization), one can increase the possibilities and conditions for the microbial inoculant to be successful [152,153].

Undoubtedly, biological control must be adapted to the changes in agriculture derived from climate change. The integration of actions within the ecological domain will require the development of new basic and applied research, carried out by multidisciplinary groups within agronomy, genetics, biological control, digital technologies, and socio-economic sciences, in which the numerous synergistic effects that will increase the sustainability of agriculture are analyzed, not only individual components, supporting the development and establishment of public policies and private initiatives for the transition toward resilient and sustainable agricultural systems [154,155].

## 5. Conclusions

As current agricultural practices are no longer sustainable and global climate change poses a great ongoing threat to agricultural yields, it is of vital importance to develop new, effective, and environmentally friendly alternatives to maintain or even increase wheat production yields, considering present and future scenarios. A promising strategy is the implementation of bioinoculants to favor plant resilience to abiotic stresses and its resistance to diseases. However, bioinoculant development must take into account critical steps to guarantee the efficacity of the final product in the field. For instance, it is essential to investigate the diverse interaction mechanisms between plants, pathogens, and beneficial microorganisms to optimize the fighting strategy and develop more efficient bioinculants. Also, it is important to take into account possible interactions of the generated bioinoculant with the native microbiome of the crop to optimize the formulation and ensure the maximum chances of survival and establishment in the intended niche. The elucidation of diverse biocontrol mechanisms has been facilitated by the implementation of omic sciences; however, there is still a general lack of knowledge, so it is essential to keep researching using different approaches. Correct bioprospection is also a main concern as not all microorganisms are adapted to all climatic conditions. In fact, due to the predicted climatic variations, there is a growing interest in employing microorganisms from extreme

environments, which exhibit higher probabilities of survival. To significantly increase the survival rates of bioinoculants in the field, they must be correctly bioformulated according to their intended application. Thus, the field application of bioinoculants is still a frontier in science that requires a large amount of research, development, and optimization, taking into account the predicted climate changes that will affect agriculture. Finally, extensive efforts must be made to transfer the diverse bioinoculant technologies to local farmers to provide them with the tools to decrease the use of chemical additives.

**Author Contributions:** Conceptualization, I.C.-A., A.C.M.-M. and S.d.l.S.V.; writing—original draft preparation, I.C.-A., A.C.M.-M., E.D.V.-R., V.V.-R. and M.A.Z.; writing—review and editing, I.C.-A., A.C.M.-M., E.D.V.-R., V.V.-R., M.A.Z., F.I.P.-C. and S.d.l.S.V. All authors have read and agreed to the published version of the manuscript.

**Funding:** This work received funding from the Instituto Tecnológico de Sonora under the PRO-FAPI program code: 2023_029 ICA. Also, funding was received from the Consejo, Nacional de Humanidades, Ciencias y Tecnologías (CONAHCYT) through the following scholarships: I.C.-A., CVU: 840486, application number: 3813573; A.C.M-M, CVU: 440879, application number: 2306476; and V.V.-R. CVU: 924892, application number: 712969. M.A.Z. CVU: 893199.

**Institutional Review Board Statement:** Not applicable.

**Informed Consent Statement:** Not applicable.

**Data Availability Statement:** Not applicable.

**Acknowledgments:** The authors would like to thank the Laboratorio de Biotecnología del Recurso Microbiano (LBRM), as well as the graphic design team for the various figures presented in this review (Abigail Reyna Luis and Sandra Guadalupe Romero Silva).

**Conflicts of Interest:** The authors declare no conflict of interest.

## Abbreviations

The following abbreviations are used in this manuscript:

| | |
|---|---|
| BCAs | Biological control agents |
| BGC | Biosynthetic gene clusters |
| CC | Coiled-coil |
| CFSs | Cell-free supernatants |
| CFU | Colony-forming units |
| ET | Ethylene |
| ETI | Effector-triggered immunity |
| ISR | Induced systemic resistance |
| JA | Jasmonic acid |
| MAMPs | Microbial-associated molecular patterns |
| MAPKs | Mitogen-activated protein kinases |
| MTI | Microbe-triggered immunity |
| NB-LRR | Nucleotide-binding and leucine-rich repeat |
| PAMPs | Pathogen-associated molecular patterns |
| PGPB | Plant growth-promoting bacterium |
| PR | Pathogenesis-related |
| PRRs | Pathogen recognition receptors |
| PTI | Pattern-triggered immunity |
| ROS | Reactive oxygen species |
| SA | Salicylic acid |
| SAR | Systemic acquired resistance |
| SNP | Single-nucleotide polymorphism |
| TIR | Interleukin-1 receptor-like |
| UV | Ultraviolet |
| WDG | Water-dispersible granules |
| WGs | Wettable granules |
| WPs | Wettable powders |

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
