# Peer review of "The Mitigation of Phytopathogens in Wheat under Current and Future Climate Change Scenarios: Next-Generation Microbial Inoculants"

_sustainability, doi:10.3390/su152115250_

Round 1

Reviewer 1 Report

The manuscript by Campos-Avelar et al. reviewed recent knowledge about crop/wheat-microbes interaction and bio-control of phytopathogens. Overall the manuscript is well structured and the introduction is comprehensive. The author also introduced many recently published findings about the advances of multi-omics and application of the approaches to predict plant/wheat-microbes interaction, which may provide guidance for improving wheat protection and production. I mainly have one comment. The author introduced a lot characters/findings about plant as general (especially in section 2). Do these findings of plant apply to wheat? Since this review should have a focus on wheat, can you modify the text/information to strengthen the topic?

Some other comments/suggestions:

Title: what do you mean by “current scenario”? Climate change is currently happening, especially in recent years.

Line 427. Change health to healthy

It’s better to make a list of abbreviations used in the manuscript. As the manuscript has lots of abbreviations and the readers may get confused. Also note that some abbreviations were defined multiple times, such as SAR (Line134, 229, 356), BCA (Line 423, 431 443). SAR also corresponds to different names, Systemic Acquired Resistance and SA-dependent systemic responses. Please check all the abbreviations again.

Line576. What is essential? Subject of the sentence is missing.

Line854, 856, 861. Subject is missing before the “is”. Can be “It is”

Author Response

Dear Reviewer, 

We appreciate all your suggestions, which were attended.

Regards,

Reviewer 2 Report

The further corrections have to be made:

Line 301: insert "affect the" after "negatively".

Lines 480 to 486: a very huge sentence without clear sense. Delete it or repair.

Line 487: insert "and" after "interest".

Line 745: insert "it" before "is".

Line 854: insert "it" before "is".

Line 856: insert "it" before "is".

Line 861: insert "it" before "is".

The further corrections have to be made:

Line 301: insert "affect the" after "negatively".

Lines 480 to 486: a very huge sentence without clear sense. Delete it or repair.

Line 487: insert "and" after "interest".

Line 745: insert "it" before "is".

Line 854: insert "it" before "is".

Line 856: insert "it" before "is".

Line 861: insert "it" before "is".

Author Response

(The authors gave the same response as above.)

Reviewer 3 Report

What is the main question addressed by research?

The central focus of this research is finding ways to reduce the impact of phytopathogens on wheat, especially in the context of current conditions and anticipated climate change scenarios. The study investigates the potential of advanced microbial inoculants to enhance crop resilience and combat phytopathogens, ultimately aiming for sustainable wheat production.

Originality and Relevance:

The subject area is highly pertinent and addresses a crucial gap in agricultural science, specifically concerning the effects of climate change on plant health. The novel approach of utilizing next-generation microbial inoculants to mitigate these effects is emphasized. The research offers a new perspective and innovative strategies to protect wheat crops from evolving phytopathogenic threats under changing environmental conditions.

Contribution to the Subject Area:

This research significantly advances the subject area by thoroughly evaluating next-generation microbial inoculants as a sustainable solution to combat phytopathogens in wheat. The review effectively consolidates existing knowledge, emphasizing the potential of these inoculants to enhance wheat crop health. This fills a significant gap in the existing literature.

Methodology and Further Controls:

Additionally, discussing potential biases or limitations in the reviewed studies would enhance the overall credibility of the assessment. The review provided highly specialized content, encompassing all relevant and specific information related to the given topic. The comprehensive contents of the review were meticulously gathered to ensure a thorough and detailed exploration of the subject matter.

Consistency of Conclusions:

The conclusions drawn align seamlessly with the presented evidence and arguments throughout the review. The authors effectively link the potential of next-generation microbial inoculants to mitigating phytopathogens, which is in line with the primary research question. The conclusions are substantiated by the reviewed literature and provide a clear direction for future research and applications in agricultural practices.

Appropriateness of References:

The references included are pertinent and encompass a comprehensive range of research articles, reviews, and studies, providing a well-rounded foundation for the review's arguments and conclusions.

In addition, there is a need to follow the guidelines for references cited in the text based on MDPI guidelines.

Comments on Tables and Figures:

The inclusion of tables and figures in the text, which summarized the main findings and depicted the mechanisms of action of microbial inoculants in mitigating phytopathogens, significantly amplified the impact and improved clarity for readers.

Table 2 authors own; if not, provide the proper citations.

The English language is acceptable.

In summary, the review on "Mitigation of Phytopathogens in Wheat under Current and Climate Change Scenarios: Next-generation Microbial Inoculants" effectively addresses the primary research question and offers valuable insights into an emerging and relevant field. It makes a noteworthy contribution by consolidating knowledge and providing a basis for further research and advancements in agricultural practices. However, augmenting the critique of methodology, incorporating visual aids, and engaging in a deeper discussion on controls would further fortify the review's impact and comprehensiveness.

The English language is acceptable. Need more originality of English language. 

Author Response

(The authors gave the same response as above.)

Reviewer 4 Report

I have reviewed the manuscript, and here are the comments:-

English needs to improve; there are some sentences and parts of the manuscript that are difficult to understand

Many abbreviations in the manuscript need to be cleared, and the aim of the study must be clear.

Please arrange all keywords in alphabetical order.

Section: 2.2.1 Population dynamics of naturally occurring phytopathogens, check, and rewrite

To improve the quality of the paper, update the reference list by adding 2022 and 2023 references.

Enhance the resolution of Figures 1, and 2.

The references section needs critical revision. The majority of the journal names are NOT capitalized as below. References that contain many authors must be shortened according to the journal format; Please correct them.

In addition, many scientific names in the references must be written in italic format.

Please follow the authors' instructions on how they write the references in the list. Why are you capitalizing the first letter of every word? Please see the journal style. For references about textbooks, please add the page numbers of the textbook. Also, please add the city of the publisher.

English needs to improve; there are some sentences and parts of the manuscript that are difficult to understand

Author Response

(The authors gave the same response as above.)

Round 2

Reviewer 4 Report

The authors did all my corrections and the paper can now be accepted in its current format.